# Fishers' Decisions to Adopt Adaptation Strategies and Expectations for Their Children to Pursue the Same Profession in Chumphon Province, Thailand

**Sukanya Sereenonchai and Noppol Arunrat ***

Faculty of Environment and Resource Studies, Mahidol University, Nakhon Pathom 73170, Thailand;
ssereenonchai@gmail.com
* Correspondence: noppol.aru@mahidol.ac.th

**Abstract:** Coastal communities and small-scale fisheries are highly vulnerable to climate change. In this study, we aimed to examine fishers' decisions to adapt to climate change and their expectations for their children to pursue the same profession. Data were obtained from fisher households covering 8 districts and 22 sub-districts in the coastal area of Chumphon Province, Thailand, using participatory observation, focus group discussion, and in-person field surveys. A binary logistic regression model was used to determine factors influencing the fishers' decisions and their expectations for their children to inherit their occupation. Results showed that the fishers are aware of the increasing trends in air temperature, sea water temperature, inland precipitation, offshore precipitation, and storms. Increased fishing experience and fishing income increased the likelihood of the fishers applying adaptations to climate change. Looking to the future, fishers with high fishing incomes expect their children to pursue the occupation, whereas increased fishing experience, non-fishing incomes, and perceptions of storms likely discourage them from expecting their children to be fishers. Of the fishers interviewed, 58.06% decided to apply adaptations in response to climate change by incorporating climate-smart agriculture, particularly by cultivating rubber, oil palm, and orchards as a second income source. The adoption of climate-smart fisheries should be considered in relation to the body of local knowledge, as well as the needs and priorities of the fisher community. To cope with the impacts of current and future climate change on coastal communities, the national focal point of adaptation should be climate change, and related governmental agencies should pay more attention to these key factors for adaptation.

**Keywords:** fisher; climate change; adaptation; expectation; Thailand

## 1. Introduction

Coastal regions around the globe are dynamic, and they are vulnerable to climate change impacts that result from the density of the population and development projects in near coastal areas. This means that people living in coastal areas directly experience the effects of climate change, such as sea level rise and monsoons [1], which means they are more likely to be convinced about the impacts of climate change than those who do not directly suffer from the consequences. These climate change impacts are predicted to force fisher villages around coastal areas to adapt in response to the changing conditions [2], and an adaptation strategy will be required based on data collected from the fisher communities, including their knowledge, skills and experience, and socio-economic context [3].

The fisheries sector provides job opportunities that promote the local economy [4]. Located in the southern part of Thailand, Chumphon Province is at considerable risk of suffering from climate change impacts, such as the tropical cyclones that usually arrive between October and December

and cause losses. Chumphon has a long coastal area along the Gulf of Thailand that increases its vulnerability to cyclones and subsequent damage. Typically, Chumphon is exposed to weather events such as strong winds and heavy rain, often resulting in huge floods. Typhoon Gay hit the Pa Tew and Tha Sae Districts on 4 November 1989 and caused tremendous losses, it being the only typhoon to have ever hit Thailand. According to the recorded data by the Thai Meteorological Department [5], seven tropical cyclones arrived at Chumphon between 1951 and 2016: one typhoon, one tropical storm, and five tropical depressions. One of them arrived in May (2007), four in November (1963, 1989, 1996, and 1998) and two in December (1972 and 2006). In January 2019, Tropical Storm Pabuk buffeted coastal villages on Southern Thailand's east coast [6]. Ferrol–Schulte et al. [7], Sales [8], and Salik [9] reported that low-income fishers are likely to suffer the most from the impacts of climate change. This has increased the focus on the impacts of climate change on small-scale coastal fisheries and coastal communities [10,11]. The ability of coastal fishers to be successful in handling the pressure from climate change depends on their ability to adapt in response to the challenges [12–14], but little is known about such adaptation strategies in Thailand. Before any adaptation measures can be designed, information is needed on what fishers think is the climatic feature that most impacts their lives. The implementation of policies might lead to more adaptive capabilities for small-scale fishers. An understanding of the local context can improve adaptation because different communities differently respond to climate change. To the best of our knowledge, no studies have been completed in Thailand on fishers' decisions to adopt adaptation strategies, nor on the expectation of their children to pursue the same employment. Transitioning to climate-smart strategies to build adaptive capacity is urgently needed in this area. These factors led to the research focus questions: Which factors influence fishers' decisions to adapt to climate change? Which factors influence their expectations that their children should inherit their occupation and become fishers? Therefore, we aimed to fill these gaps and address the research questions for more effective climate adaptation policy improvements, especially given the national focal point of adaptation to climate change, which is operated by the related governmental agencies.

## 2. Methodology

### 2.1. Study Area

Chumphon Province is located in the upper southern part of the west side of Thailand, between 9.6° N and 11.0° N and between 98.7° E and 99.5° E, and 463 km from Bangkok. Chumphon Province has a tropical monsoon climate with two seasonal monsoons: the southwest monsoon that blows in from the Indian Ocean from May to mid-October, bringing moisture from the sea and potential rainfall to Thailand, and the northeast monsoon that blows in from the northeasterly direction, bringing cooler temperatures, less humidity, and occasional rainfall to Thailand from mid-October to mid-February, especially in October and November. On average per year, the temperature is 27 °C, the highest temperature is 31.9 °C, and the lowest temperature is 23.5 °C. April is the hottest month of the year, at 38.8 °C. The precipitation rate is 1827.7 mm per year, and it rains an average of 168 days per year. November is the month with the most rainy days, with an average of 287.9 mm of rain and the highest record of rainfall within 24 h was 423.5 mm [5].

### 2.2. Data Collection

Data were obtained from participatory observations, focus group discussions, and in-person field surveys. The sample of fisher households covered 8 districts and 22 sub-districts in the coastal area (Figure 1). Stratified random sampling was used in each sub-district to select the sampling villages. Then within each village, five to eight fisher households were randomly selected. As a result, the total survey sample included 89 villages and 528 fisher households. The fishers were randomly selected from those with fishery experience of at least 3 years. To answer the main research questions, the following main points were the focus:

(1)   Fishers' perceptions about climate change. The fishers were asked about their perceptions of climate change during the past 10 years by provoking reflection back to their practices and transitions in fishing.

(2)   Adaptation strategies and climate-smart strategies to build adaptive capacity. The fishers were asked about their strategies for coping with climate change during the past 10 years to understand their adoption behavior.

(3)   Expectation and reasons for their children to be fishers;

(4)   Socio-demographic characteristics (sex, birth place, fishing experience, education level, and household size).

(5)   Income including fishing and non-fishing income.

(6)   Perspectives on climate variability, including perception of offshore precipitation, inland precipitation, air temperature, sea water temperature, saline water intrusion, and storms.

(7)   Institutional accessibility, including assistance from institutions and social capital.

(8)   Communicating adaption to climate change (CACC), including sender, message, channel, and receiver.

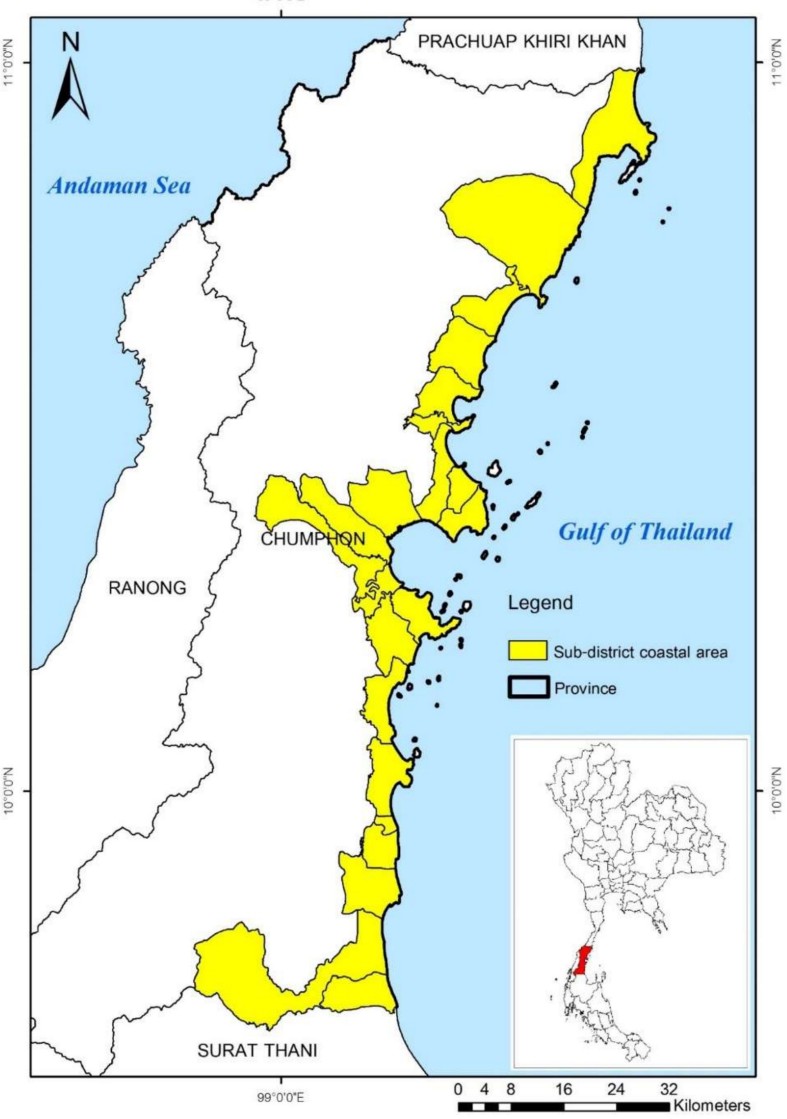

**Figure 1.** Study area.

*2.3. Empirical Model*

The binary logistic regression model was used to determine the factors influencing the fishers' decisions to adapt to climate change (yes or no) and the fishers' expectations for their children to inherit their occupation (yes or no). Each fisher's decision is represented by a binary variable ($y = 0$ or $y = 1$). The explanatory variables are defined as either dichotomous or continuous variables, as detailed in Table 1. According to the Bernoulli probability function, as the value of 1 is the probability $P_i$ and 0 is the probability $1 - P_i$, the odds of the event occurring are $P_i/1 - P_i$. The Bernoulli probability function can be expressed as:

$$\ln\left[\frac{P_i[Y_i = 1]}{1 - P_i[Y_i = 1]}\right] = \ln(Odds) = \propto_0 + \sum_{k=1}^{k} \beta_k X_{ik} \tag{1}$$

$$Odds = \left[\frac{P_i[Y_i = 1]}{1 - P_i[Y_i = 1]}\right] = exp\left[\propto_0 + \sum_{k=1}^{k} \beta_k X_{ik}\right] \tag{2}$$

$$\begin{aligned} y = &\propto_0 + \beta_1 X_1 + \beta_2 X_2 + \beta_3 X_3 + \beta_4 X_4 + \beta_5 X_5 + \beta_6 X_6 + \beta_7 X_7 + \beta_8 X_8 + \beta_9 X_9 + \\ &\beta_{10} X_{10} + \beta_{11} X_{11} + \beta_{12} X_{12} + \beta_{13} X_{13} + \beta_{14} X_{14} + \beta_{15} X_{15} + \beta_{16} X_{16} + \beta_{17} X_{17} + \\ &\beta_{18} X_{18} + \beta_{19} X_{19} + \varepsilon \end{aligned} \tag{3}$$

**Table 1.** Explanatory variables in the binary logistic regression model.

| Variable | Description |
| --- | --- |
| **Socio-Demographic Characteristics** | |
| Gender ($X_1$) | Dummy, 1 if the household head is male; 0 otherwise |
| Birth place ($X_2$) | Dummy, 1 if the fisher is native; 0 if the fisher is immigrant |
| Fishing experience ($X_3$) | Continuous, number of years fisher has worked related to fishery |
| Schooling ($X_4$) | Continuous, number of years in school completed by fisher |
| Household size ($X_5$) | Continuous, number of household members |
| **Household Income** | |
| Fishing income ($X_6$) | Continuous, direct total income from fishing in Baht per month |
| Non-fishing income ($X_7$) | Continuous, total income from other sources in Baht per month |
| **Perception of Climate Change** | |
| Perception of offshore precipitation ($X_8$) | Dummy, 1 if the fisher has perceived the increasing offshore precipitation; 0 otherwise |
| Perception of inland precipitation ($X_9$) | Dummy, 1 if the fisher has perceived the increasing inland precipitation; 0 otherwise |
| Perception of air temperature ($X_{10}$) | Dummy, 1 if the fisher has perceived the increasing air temperature; 0 otherwise |
| Perception of sea water temperature ($X_{11}$) | Dummy, 1 if the fisher has perceived the increasing sea water temperature; 0 otherwise |
| Perception of saline water intrusion ($X_{12}$) | Dummy, 1 if the fisher has perceived the increasing saline water intrusion; 0 otherwise |
| Perception of storm ($X_{13}$) | Dummy, 1 if the fisher has perceived the increasing storm; 0 otherwise |

**Table 1.** *Cont.*

| Variable | Description |
|---|---|
| **Institutional Accessibility** | |
| Assistance from institutions ($X_{14}$) | Dummy, 1 if the fisher has been assisted related to climate change impact and adaption; 0 otherwise |
| Social capital ($X_{15}$) | Dummy, 1 if the fisher has accessed to the relationship of fisher extension; 0 otherwise |
| **CACC Characteristics** | |
| Sender ($X_{16}$) | Dummy, 1 if the fisher has received information from trusted senders; 0 otherwise |
| Message ($X_{17}$) | Dummy, 1 if the fisher has received proper and visualized adaptation techniques; 0 otherwise |
| Channel ($X_{18}$) | Dummy, 1 if the fisher has received information through accessible communication channels; 0 otherwise |
| Receiver ($X_{19}$) | Dummy, 1 if the fisher is satisfied with adaptation techniques and has been convinced to adapt; 0 otherwise |

### 2.4. Climate Trend and Variability Analysis

Precipitation and temperature data from Chumphon meteorological station run by the Thai Meteorological Department from 1987 to 2016 were analyzed to validate the farmers' perceptions on precipitation and temperature changes. The trends in precipitation and temperature were analyzed to produce summaries. Linear regression was used to determine the slopes and direction of trends.

## 3. Results

### 3.1. Fishers' Perceptions and Indigenous Knowledge

The results showed that the fishers had different levels of perception of climate change impacts: believe climate change has led to increases in air temperature (82.50%), sea water temperature (58.30%), inland precipitation (45.30%), offshore precipitation (68.42%), and storms (71.60%), whereas 61.20% did not perceive any changes in saline water intrusion (Figure 2). From a comparative analysis using scientific data, we found that the perceptions of the fishers regarding temperature and precipitation were consistent with the scientific data over the past 30 years, which shows that temperature and precipitation are likely to increase by 0.02 °C·year$^{-1}$ and 4.15 mm·year$^{-1}$, respectively (Figure 3).

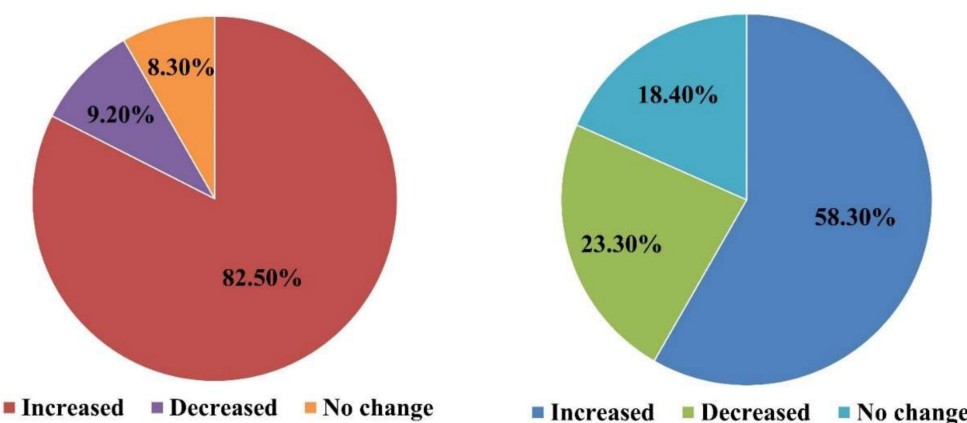

**Figure 2.** *Cont.*

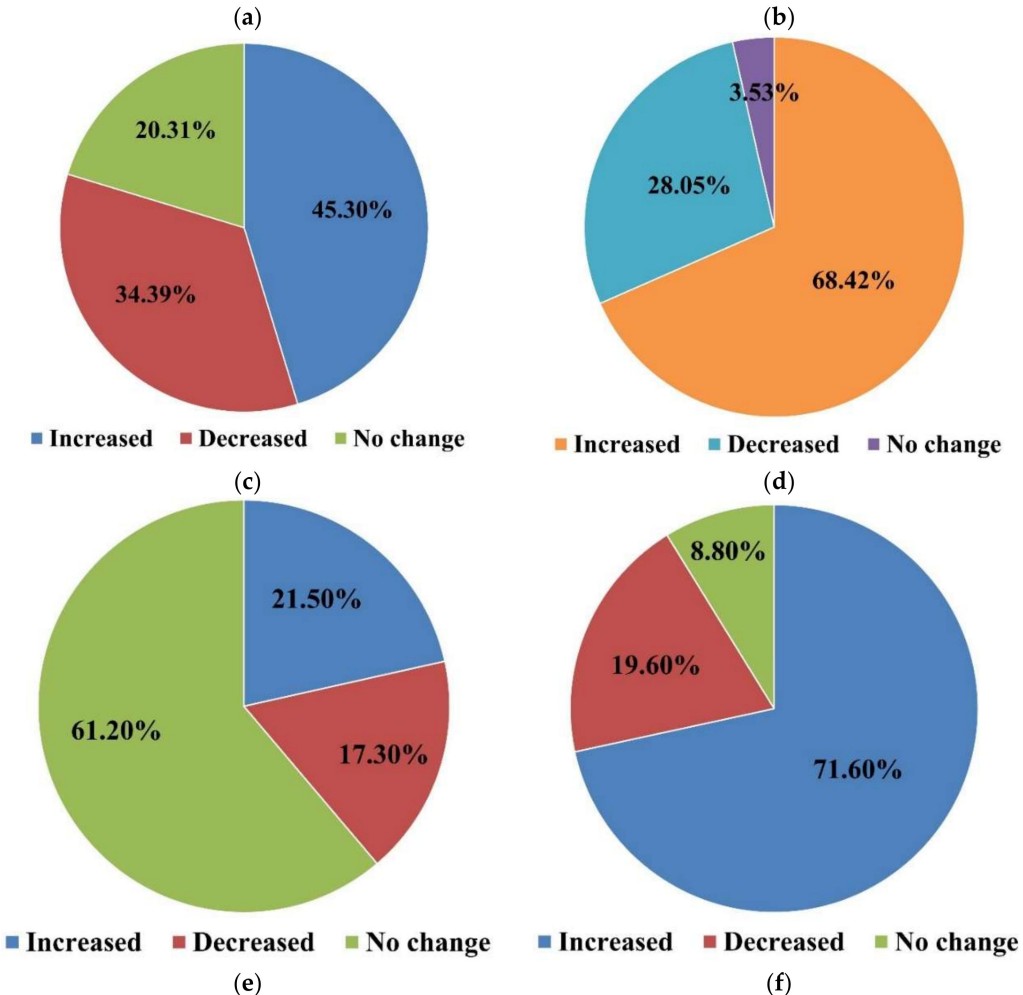

**Figure 2.** Fishers' perceptions about climate change.

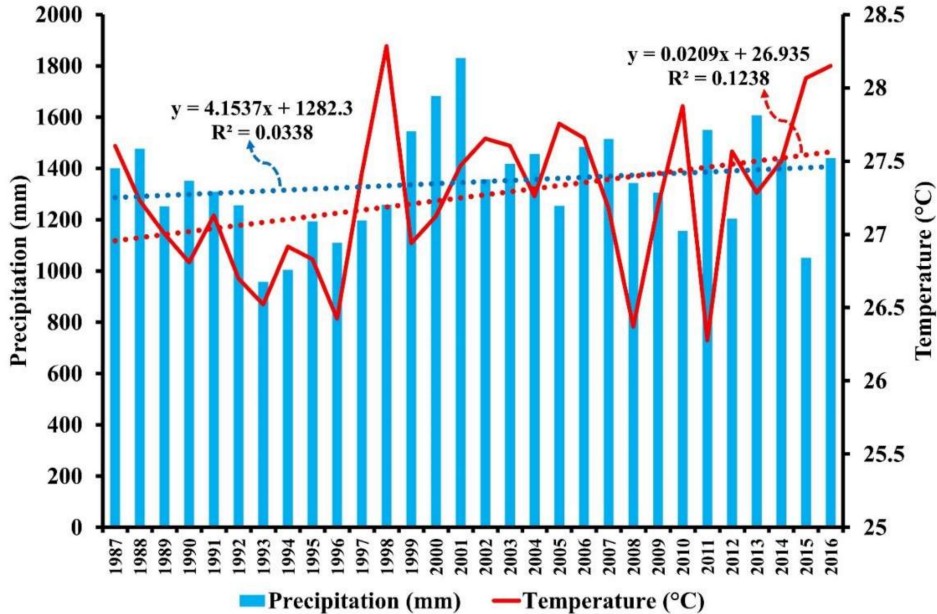

**Figure 3.** Precipitation and temperature trends over the past 30 years (1987–2016).

Based on in-depth interviews and focus group discussions with the fishers and coastal communities in Chumphon Province to understand their indigenous knowledge, we revealed that: (1) in very hot areas, aquatic animals will not mate or lay eggs, resulting in smaller numbers—especially squid and blue swimming crabs; (2) when they are sailing in the midst of the sea, if they notice any area with unclear water, they know that squids will not be found in the area; (3) if the wind keeps changing its direction back and forth, only a small number of fish will be found; (4) when fish are seen swimming still, it means that a strong wind, waves and rainfall are coming shortly; (5) when squids are not out to enjoy the light from squid boats, it signals a potential storm that night; and (6) when aquatic animals such as fish, crabs and shellfish uncharacteristically disappear, and when terrestrial animals perform certain actions, such as ants moving their eggs well above the ground, or cockroaches flying, this suggests that a storm is inbound.

### 3.2. Fishers' Decisions to Adapt to Climate Change and Expectations for Their Children

#### 3.2.1. Socio-Demographic Characteristics

Fishing experience is highly significant and positively related to climate change adaptation, followed by birthplace and education level, which also contribute to the likelihood of adoption. This indicates that native fishers who are relatively experienced do not expect their children to pursue the same occupation (Table 2).

**Table 2.** Fishers' decisions to adapt, and expectations of their children.

| Variable | Mean | Fishers' Decisions to Adapt (1 = Yes; 0 = No) | | Fishers' Expectations of Their Children (1 = Yes; 0 = No) | |
|---|---|---|---|---|---|
| | | Coefficient Estimates | Marginal Effects | Coefficient Estimates | Marginal Effects |
| **Socio-Demographic Characteristics** | | | | | |
| Gender | 0.93 | 0.680 | 0.023 | 0.39 | 0.016 |
| Birth place | 0.62 | 8.00E−05 | 7.00E−06 * | −0.00052 | −6.80E−05 ** |
| Fishing experience | 42 | 0.018 | 0.0052 *** | −0.022 | −0.0086 *** |
| Schooling | 7.0 | 0.012 | 0.006 * | −0.007 | −0.00082 |
| Household size | 5.0 | 0.018 | 0.007 | 0.021 | 0.0053 |
| **Household income** | | | | | |
| Fishing income | 11688 | 8.7E−06 | 7.5E−07 *** | 2.71E−06 | 5.7E−07 *** |
| Non-fishing income | 2620 | 6.4E−05 | 5.7E−06 ** | −1.58E−04 | −9.6E−06 *** |
| **Perception of Climate Change** | | | | | |
| Perception of offshore precipitation | 0.37 | 0.009 | 0.00015 | −0.008 | −0.00046 * |
| Perception of inland precipitation | 0.62 | 0.014 | 0.0056 * | −0.004 | −0.00074 |
| Perception of air temperature | 0.71 | 0.007 | 0.0027 | −0.012 | −0.0063 * |
| Perception of sea water temperature | 0.68 | 0.021 | 0.0082 ** | −0.015 | −0.0085 * |
| Perception of saline water intrusion | 0.32 | −0.003 | −0.0007 | −0.006 | −0.00093 |
| Perception of storm | 0.82 | 0.831 | 0.067 ** | −0.901 | −0.065 *** |
| **Institutional Accessibility** | | | | | |
| Assistance from institutions | 0.69 | 0.107 | 0.054 * | 0.411 | 0.062 ** |
| Social capital | 0.61 | 0.132 | 0.061 * | 0.406 | 0.047 * |

**Table 2.** *Cont.*

| Variable | Mean | Fishers' Decisions to Adapt (1 = Yes; 0 = No) | | Fishers' Expectations of Their Children (1 = Yes; 0 = No) | |
|---|---|---|---|---|---|
| | | Coefficient Estimates | Marginal Effects | Coefficient Estimates | Marginal Effects |
| **CACC Characteristics** | | | | | |
| Sender | 0.81 | 0.541 | 0.093 ** | 0.327 | 0.084 * |
| Message | 0.72 | 0.621 | 0.009 | 0.183 | 0.031 |
| Channel | 0.54 | 0.082 | 0.0079 * | 0.009 | 0.00064 |
| Receiver | 0.71 | 0.221 | 0.021 | 0.132 | 0.0028 |
| Constant | | −2.381 *** | | 3.012 *** | |

***, **, * Significant at 1%, 5%, and 10% probability level, respectively.

### 3.2.2. Household Income

Fishers with high fishing incomes are able to improve their ability to adapt, and they also expect their children to pursue the occupation, as increasing non-fishing incomes increases the likelihood of fishers making adaptations to climate change while also decreasing their expectation of their children following the same profession, as more income can be earned from jobs other than fishing (Table 2).

### 3.2.3. Perceptions of Climate Change

Increased perceptions of storms, sea water temperature, and inland precipitation among the fishers has convinced them to undertake more adaptations, whereas the perceptions of increased storm frequency, sea water temperature, air temperature, and offshore precipitation are discouraging them from expecting their children to be fishers (Table 2).

### 3.2.4. Institutional Accessibility

Support and assistance from the relevant local authorities can potentially increase the likelihood of the fishers adapting to climate change and expecting their children to continue the occupation (Table 2).

### 3.2.5. CACC Characteristics

A higher perceived credibility of the sender conveying messages about climate change, its impacts, and adaptation strategies, contributed to an increased likelihood of the fishers making adaptations to climate change and expecting their children to continue the profession (Table 2).

### 3.3. Fishers' Adaptation Strategies

Results from the field study showed that 58.06% of the fishers decided to undertake adaptations in response to climate change by applying climate-smart options, followed by 18.21%, 10.42%, 6.33%, 3.07%, and 1.88% of the fishers deciding to fish further away from the shore, shift fishing times and locations, labor in fisheries, labor in agriculture, and fishing for a longer time, respectively. The remaining 2.03% did not incorporate any adaptation strategies (Figure 4).

The adopted climate-smart options included the development of alternative livelihoods by changing to the agricultural sector (32.28%), particularly by cultivating rubber, oil palm, and orchards (rambutan, mangosteen and longkong, for instance) (Figure 4). They explained that with consideration of the unstable climate, monsoons, more severe and frequent storms, and declining fishing resources, having a second job in the agricultural sector was necessary, such as cultivating rubber, oil palm, or fruit trees, which do not require daily maintenance; they only need some time to look after the trees

after they have finished fishing. When the monsoon seasons came or when the fishing ban season started, they could harvest what they had grown to continue to earn an income.

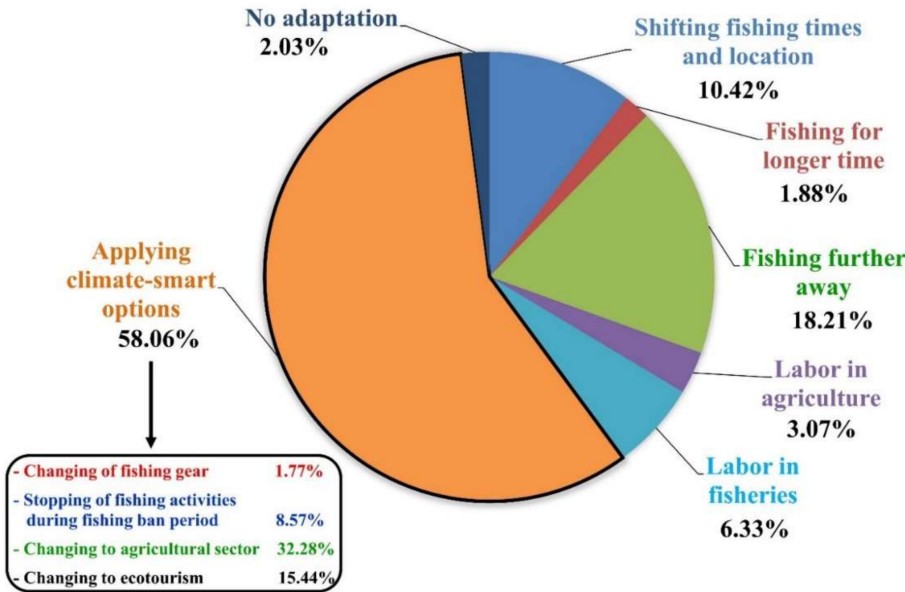

**Figure 4.** Fishers' adaptation strategies.

Of the fishers, 15.44% decided to provide ecotourism services such as homestays, restaurants, and other tourist services during tourist seasons, where they left their fishing jobs behind for a while to earn more income from ecotourism and to avoid risks from unstable currents and a limited amount of fish (Figure 4).

Of the fishers, 8.57% stopped their fishing activities during the fishing ban period and turned to join groups in their community to preserve and restore fishing resources by establishing a blue swimming crab bank and by building artificial fish habitats to help restore the natural ecology. They explained that over the past years, too many blue swimming crabs had been caught and that these activities would help to restore the blue swimming crab population and improve sustainability (Figure 4).

Of the fishers, 1.77% opted for changes in fishing gear through mechanization of the rig in terms of better engines that were more fuel-efficient (Figure 4).

We found that the fishers did not adopt these adaptation strategies one at a time. Our findings showed that 2.8% of the fishers adopted all six strategies at the same time, 12.44% adopted five, 28.43% adopted four, 12.55% adopted three, and surprisingly, 43.78% adopted only one strategy. These climate-smart options are evidence that the fishers have various methods of coping with the effects from climate change and that they are capable of selecting methods that correspond to specific situations. This means that enhancing the fishers' ability to adopt climate-smart options will lead to greater capabilities and sustainability of their profession.

It is unfortunate that relocation has been adopted by the fishers in the study area as one of the adaptation strategies. Apart from the unstable climate and declining resources, coastal erosion causing the loss of coastal lands, including the living and farming areas of the fishers, is also supporting the decision to completely stop fishing, to turn to a different occupation, or to sell the land.

## 4. Discussion

### 4.1. Fishers' Perceptions of Climate Change and Indigenous Knowledge

Fisher communities in rural areas have a long tradition of implementing indigenous knowledge systems (IKS) in handling the effects of the ever-changing climate. Indigenous or traditional knowledge

is a body of knowledge that is accumulated and passed on from generation to generation [15]. According to Nakashima et al. [16], an integration of the indigenous knowledge system and scientific research is an effective method for generating new knowledge concerning adaptation strategies at the local level, as a body of knowledge is built upon learning, identifying and applying the activities of the people within a community context [17]. Personal perception is what individuals perceive of the local climate instability, climate change and reactions, based on personal experience and values [18]. As a result, an integration of the indigenous knowledge system and personal perception enables adaptations to be implemented in response to changes in a local area, as the people have long developed familiarity with the local conditions [19]. This fact is supported by our finding which showed that the fishers in Chumphon Province used both their indigenous knowledge system and personal perception in their daily lives, and they confirmed that both are practical and reliable tools they would certainly pass on to the younger generations to enable them to cope with the impacts of climate change in the future. Their vulnerability to such effects because they relied on fishing resources, which are affected by climate change [20], is in agreement with the findings of Zhang et al. [21], who demonstrated that local fishers have an insightful understanding into the unstable climate as they typically rely on fishing resources.

### 4.2. Factors Influencing Fishers' Decisions to Adapt and Expectations of Their Children to Inherit the Occupation

Richardson et al. [22] and Dimech et al. [23] reported that, compared with migrant fishers, native or local fishers have more of an attitude that focuses on the management, preservation and restoration of fishing resources and adaptation strategies, which is in agreement with our findings demonstrating that native fishers are more active in undertaking adaptations to manage climate change and that they do not expect their children to earn a living as a fisher. They explained that the career is tiring and demanding with a low income, and catching fish is becoming more difficult due to the declining fishing resources and unstable climate, and more dangerous with increasingly severe storms. They would not like to see their children take such risks. However, this study suggests that an emphasis should be placed on younger fishers as they are not the only ones performing activities that are now in place for the management and preservation of fishing resources in the near future, but they are also the beneficiaries of these activities. This suggestion is in agreement with the study of McCook et al. [24], which suggested that the preservation of fishing resources will be more effective in the future when the focus is placed on the younger generations. As costs and benefits are the major determinants of fishers' behaviors [25], policymakers should prioritize socio-economic traits to build greater confidence in their livelihoods and to avoid conflict with rules and regulations, as well as with the community's way of living [26], despite some previous studies pointing out that adaptations of behavior toward climate change can vary significantly depending on age [21,27]. Teh et al. [27] pointed out that younger fishers are expected to be willing and eager to learn new things, and to be able to find and earn a living from different jobs as it is easier for them to learn and apply newly acquired knowledge than older fishers [28].

Fishing experience plays a role in determining a fisher's perception of climate change. More experienced fishers are less vulnerable to climate change impacts, and they are quicker to adapt [28]. This is in agreement with the study of Nhemachena and Hassan [29], which stated that the more farming experience a farmer has, the more likely the farmer will choose to accept adaptation strategies. In this case, as the fishers have earned an income over a long period of time, their experiences from learning by doing and from a variety of real-life situations allow them to react and to act against climate change effects in a timely manner, to make relatively accurate predictions, and to provide constructive advice to the younger generations.

Higher education is associated with a greater access to information about technology [30]. Lin [31] reported that education level and technology acceptance level are positively associated with the level of climate adaptation [32]. The adoption of technologies, such as telephone and radio, helps to increase

access to information that is necessary for making adaptations to climate change [33]. As a result, fishers with a higher education level are more likely to make better adaptations to climate change.

Fishing incomes are a source of support for household activities [34,35], and an increased number of fishing days can help improve income distribution when the fishers spend their fishing incomes on their household activities. This is proof that higher incomes lead to a greater awareness of climate change [29,36].

The fishers can often directly perceive climate instability, and they are likely to be more convinced about climate change than those who do not live in coastal areas. With such a perception, they are under pressure to adapt in response to climate changes. Fishers need to observe and be aware of the daily weather as part of their working routines, which is in agreement with the finding of this study that, as the fishers are well aware of climate change, they are more prepared to make adaptations. A qualitative analysis, however, has shown that most of the highly educated fishers (young fishers) demonstrated a low level of perception of climate change. This may result from the fact that they have other sources of income than fishing, which lessens their experience of climate change impacts; this is in agreement with the study of Tologbonse et al. [37], which revealed the same finding for farmers in Nigeria.

Support and assistance from local authorities were found to affect the fishers' decisions to make adaptations, as well as their expectations for their children to be fishers, which is in agreement with the study of Gbetibuou [38], who stated that access to extension services is key to the adoption of adaptation strategies. This access to extension services is positively associated with the adoption of new technologies for adaptation because access to new information and skills leads to a higher level of personal technology acceptance. In addition, we found that the more access the fishers had to extension services, the more likely they would perceive the existence of climate change. This is because such access to the information provided by extension officers enables a greater awareness of the climate situation and their fishing activities. Basic support and assistance from local authorities that help facilitate the adaptations of the fishers mostly includes the provision of physical infrastructure, especially regarding the credit system that enables the fishers to purchase tools that are necessary for their fishing activities [39]. In terms of adaptation, however, Arunrat et al. [40] analyzed debt and net income ratios and noted that small-scale farmers are more likely to have a higher ratio of debt to net income than large-scale farmers due to easy access to loans, which has placed many of them in chronic poverty. In contrast, large-scale farmers are more capable of paying debts, which subjects them only to transient poverty. As a result, the relevant authorities should not only provide fishers with access to loans and credits, but also ensure that they are well-educated about effective risk management, which is a necessity. Democracy [41] and corruption [42] affect fisheries' governance in terms of inequitable policy support and law enforcement. Therefore, suppression of corruption and reducing inequality are required.

Social capital includes the number of members in the fishing family, the expansion of a fisher network, and informal institutions, as well as private social networks, which play a pivotal role at present, as they provide fishers greater access to modern technologies, innovations, and ideas, while serving as platforms for breaking news and updates about climate. Access to social capital then helps reduce vulnerability [43], especially in stressful times, whereas a good relationship among family members, within a community, or across communities, can contribute to immediate actions being taken in response to climate change [44,45]. This social capital helps build resistance to climate change impact and increases preparedness to make adaptations, which deserves serious attention from the local authorities [46,47]. To summarize, such access to extension services will enable fishers to undertake effective adaptations to handle climate change. There is hope that social capital will continue improving, as well as technology becoming more advanced and effective, so that the expectations of the fishers for their children becoming fishers will persist.

Effective adaptation communication is a process of engagement, requiring collaboration, understanding, and patience to achieve the desired outcome. The success factors for communicating

adaptation consist of (1) trusted senders and opinion leaders inside the community; (2) content and messages should be explained and visualized in understandable ways, and the local people should become more familiar with clear implications for actions at different times, as well as adaptation languages, and positive terms, explained through concrete actions, should be related to responsible and high-priority planning, decision-making, and management of uncertainty; (3) messages and media should be framed, designed, and employed based on audiences or target groups: framing issues using communication formats and channels that are appropriate for the target group; and (4) receivers should be convinced regarding adaptation strategies [48–51].

Even though the findings of this study only point to the influence of the sender's credibility on decisions to undertake adaptations against climate change, factors, such as the message, channel, and receiver, all play an important roles in decision making. The sender serves not only as a distributor of knowledge and information, but also as a fellow villager living in the same context with the community who shares a good understanding of the way of life in the fisher village and the social capital. If the sender has experience in making adaptations in response to the climate, they will also be able to serve as a role model for fellow fishers from whom to learn, as they seek to continue living as fishers in the face of climate change impacts. A sender is thus important for distributing knowledge, convincing the fishers, and promoting adaptation behaviors. Adger [52] added that the ability to make adaptations is directly associated with social capital, which can be increased through a relationship that is built on trust, networking, reciprocation, willingness, and a group ability to do activities together.

### 4.3. Fishers' Adaptation Strategies

The fishers' decisions to change the time and place for their fishing activities depended on the opportunities that they have noticed, and this kind of decision to adapt often leads to increased incomes, saved costs, and lower risks due to unstable weather.

An adoption of the new engine that consumes energy more effectively is an adoption of green technology, which is costly at first. The fishers are well aware of the costs and benefits of such an adoption, knowing that the technology will help them save costs and improve their performance, while decreasing their need for labor. This is in agreement with the findings reported by Chali et al. [53], which stated that 19% of rigs in Lake Kariba, Zambia are equipped with fish finders.

Positive adaptation strategies, particularly an investment of non-fishing incomes, should be promoted, as they have been proven to be necessities for the fishers as they will help build greater resilience against climate change impacts [47,54].

Indigenous knowledge about adaptation strategies, collected through learning by doing and through real-world applications by a community, will help enhance such strategies and increase their effectiveness [55].

Keys et al. [56] stated that the ability to make adaptations can be enhanced through three major strategies: the development of knowledge and understanding about climate change, school adaptation, and the community's network development, and the participation of the villagers in identifying sources of specific risks.

Relocation is a process that often results in a poor ability to make adaptations in the face of serious environmental changes [44,57]. However, from a wider perspective, relocation can lead to the improved ability of a community to make adaptations to handle climate change, as relocation has prepared them for any changes [58].

Our study has shown that small-scale fishers are a source of basic knowledge about the environment, aquatic ecosystem, fish, and other aquatic animals, as well as the local climate. This means that an adoption of climate-smart fisheries should be considered in relation to the body of knowledge about fisheries in general, as well as the needs and priorities of the fisher community. The focus of the adoption of climate-smart strategies, as a result, should be placed on increasing productivity in a sustainable fashion, improving resilience, mitigating greenhouse gas (GHG) emissions, reducing the vulnerability to climate change impacts, and creating a more

sustainable way of living. To determine what climate-smart strategies can be useful and practical to a fisher community, the participation of the villagers is crucial. The climate-smart approaches to be specified by policymakers should be well-known to the community and should fit within the local context, so that the operational staff can fully implement them and so that the maximum benefit is reached for the beneficiaries at all levels.

## 5. Conclusions

This study contributes to the understanding of fishers' decisions to adopt adaptation strategies, as well as their expectations for their children to pursue the same occupation. Transitioning climate-smart strategies to build adaptive capacity is urgently required for Chumphon Province, Thailand. Our results showed that the fishers have different perceptions of climate change impacts, with most noting increases in air temperature, sea water temperature, inland precipitation, offshore precipitation, and storms, whereas the majority do not perceive any changes in saline water intrusion. Increasing fishing experience and fishing income raised the likelihood of the fishers to undertake adaptations to climate change. In the future, fishers with high fishing incomes expect their children to pursue the occupation, whereas increased fishing experience, non-fishing incomes, and perceptions of storms likely discourage them from expecting their children to be fishers. A total of 58.06% of the fishers have implemented adaptations in response to climate change by applying climate-smart agriculture, particularly by cultivating rubber, oil palm and orchards, because they must have a second job in the agricultural sector. An adoption of climate-smart fisheries should be considered in relation to the body of local knowledge as well as the needs and priorities of the fisher community. Therefore, the national focal point and related agencies should pay more attention to these key factors for adaptation to cope with current and future climate change.

**Author Contributions:** All authors have contributed equally to conceptualization, analysis, writing, and investigations.

**Funding:** This research project was supported by Mahidol University for the fiscal year of 2017, grant number 860150013000.

**Acknowledgments:** The authors would like to thank Thai Meteorological Department for providing the climate data. Furthermore, the authors are grateful to the reviewers for their helpful comments to improve the manuscript.

**Conflicts of Interest:** The authors declare no conflict of interest.

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
