# Peer review of "Fishers’ Decisions to Adopt Adaptation Strategies and Expectations for Their Children to Pursue the Same Profession in Chumphon Province, Thailand"

_climate, doi:10.3390/cli7020034_

Round 1

Reviewer 1 Report

- The manuscript makes a good description of fishermen´s communities and tries to analyze the perception of fishermen regarding climate change adaptations and perceptions whether their children will inherit the same job as fishermen.

- there is no hypotheses or research question.

- the methods for data aquisition are ok, but we really do not know how many interviews or field survey the authors performed. The sampling period is too short for talking about perceptions.- The data obtained during the field surveys is very usefull to describe but is nor enough to understand fishermen´s behaviour, like for example the adoption or not of climate-smart fishing strategies........or what to expect with the future of their children, if they will be fishermen as well or not

- the way the results are presented did not help to clarify the aims of the study. the tables are confussing and the pie charters did not contribute too much.

- the english language needs some extra editing.

Author Response

Dear Editor and reviewer,

Thank you very much for your kind efforts in reviewing our manuscript.

We greatly appreciate the valuable and constructive comments from the reviewers. Along the lines suggested by the reviewers, we have made comprehensive revisions and have significantly improved the writing of the manuscript. In our responses to the reviewers below, details of the changes we have made in the manuscript can be found orderly.

We hope that the revised manuscript can reach the criteria for publication in Climate.

Yours sincerely,

The authors

Reviewer: 1

Comment:

- The manuscript makes a   good description of fishermen´s communities and tries to analyze the   perception of fishermen regarding climate change adaptations and perceptions whether   their children will inherit the same job as fishermen.

- there is no hypotheses   or research question.

Response:

We have already   identified the research   question as shown in Lines 70-72.

       “These lead to the research questions   that what are the influence factors for fisherman’s decision to adapt to   climate change, and to expect their children to inherit and be the   fisherman?”

Comment:

- the methods for data   aquisition are ok, but we really do not know how many interviews or field   survey the authors performed. The sampling period is too short for talking   about perceptions.- The data obtained during the field surveys is very   usefull to describe but is nor enough to understand fishermen´s behaviour,   like for example the adoption or not of climate-smart fishing   strategies........or what to expect with the future of their children, if   they will be fishermen as well or not

Response:

We have added more information about the number   of interviews as shown in Lines 98-99.

   “As a result, the total survey samples included 89 villages and   528 fisherman households.

To understand fishermen’s behaviour, their perceptions on climate change, and adaptation strategies, we asked the fishermen to think back in the   past 10 years. We have already mentioned in Lines 101-106.

“(a) Fishermen’ perception about climate change: The fishermen were asked   about their perception on climate change during the past 10 years by   thought-provoking back to their practice and transition on fishing.  

(b) Adaptation strategies and climate smart strategies to build adaptive capacity: the   fishermen were asked about their strategies to cope with climate change   during the past 10 years and to understand their adoption behavior

Comment:

- the way the results are   presented did not help to clarify the aims of the study. the tables are   confussing and the pie charters did not contribute too much.

Response:

We have already improved Fig.4. Table 2 was explained in section 3.2   Fishermen’s decision to adapt and expectation on their children.

Comment:

- the english language   needs some extra editing.

Response:

This manuscript was edited by   Proof-Reading-Service.com.

Reviewer 2 Report

I have read the manuscript ”Fishermen’s decision to adopt adaptation strategies and expectation
on their children to pursue the same walk of life in Chumphon Province, Thailand”

I think this might be a valuable case study and can see this work being published. I think the author(s) deserve accolades for taking on such important themes of research and doing so by conducting original field work.

I do not study fishermen per se, so my comments are more on structure of this article.

Some critical comments, that may be seen as constructive ways to further improve this text:

-       State the research question early on, even in the introduction.

-       The piece seems to have two aims, one on strategies of adaptation and one on views on whether children will inherit the trade. Can a section that explains the choice of having both these two aims do them more connected? As it stands now, it almost feels like the theme for separate papers, though I see the link.

-        

One substantial reflection: Can the piece engage more in describing the governance of Thailand? I suppose there is a debate on whether the degree of democracy affects resource management (Jagers et al. in Ocean & Coastal Management) and how the degree of corruption affects perceptions of fisheries governance (Sundström in Global Environmental Change). This can be added to page 17, where government assistance is discussed. It is likely that many fishermen view Thai government officials as untrustworthy, though this is not particularly discussed.

In general, the study does not seem to have a typical theory section that outline expectations or motivate the specific aim of the study. More constructively, I think it would be good to discuss such studies as mentioned above, because these factors such as democracy and corruption will likely prove important in order to craft effective adaptation strategies in the future for fisheries.

Minor changes relates to grammar and I recommend a thorough reading of the English. See, e.g., page 15, the heading 4.2. should be Factors, with an s, I suppose. Should heading 4 be Discussion rather than Discussions?

Some sentences need rephrasing, such as – on page 3 – “… To answer the main research questions on:”

Author Response

Dear Editor and reviewer,

Thank you very much for your kind efforts in reviewing our manuscript.

We greatly appreciate the valuable and constructive comments from the reviewers. Along the lines suggested by the reviewers, we have made comprehensive revisions and have significantly improved the writing of the manuscript. In our responses to the reviewers below, details of the changes we have made in the manuscript can be found orderly.

We hope that the revised manuscript can reach the criteria for publication in Climate.

Yours sincerely,

The authors

Reviewer: 2

Comment:

I have read the manuscript   ”Fishermen’s decision to adopt adaptation strategies and expectation on their   children to pursue the same walk of life in Chumphon Province, Thailand”

I think this might be a   valuable case study and can see this work being published. I think the   author(s) deserve accolades for taking on such important themes of research   and doing so by conducting original field work.

I do not study fishermen   per se, so my comments are more on structure of this article.

Some critical comments,   that may be seen as constructive ways to further improve this text:

-       State the research question early on,   even in the introduction.

Response:

We have already   identified the research   question as shown in Lines 70-72.

       “These lead to the research questions   that what are the influence factors for fisherman’s decision to adapt to   climate change, and to expect their children to inherit and be the   fisherman?”

Comment:

-       The piece seems to have two aims, one   on strategies of adaptation and one on views on whether children will inherit   the trade. Can a section that explains the choice of having both these two   aims do them more connected? As it stands now, it almost feels like the theme   for separate papers, though I see the link.

Response:

     In   the context of adaptation strategies, people can reflect the current and   future perspectives because if they have been adopted, they are confident on   their potential to do so.  Therefore,   they can propose their intentions and expectations on their children to   pursue the same walk of life.

   This is   why the two main aims can be connected together and can put the readers to   clearly understand not only the present situation, but also get the future   views.

Comment:

One substantial   reflection: Can the piece engage more in describing the governance of   Thailand? I suppose there is a debate on whether the degree of democracy   affects resource management (Jagers et al. in Ocean & Coastal Management)   and how the degree of corruption affects perceptions of fisheries governance   (Sundström in Global Environmental Change). This can be added to page 17,   where government assistance is discussed. It is likely that many fishermen   view Thai government officials as untrustworthy, though this is not   particularly discussed.

In general, the study does   not seem to have a typical theory section that outline expectations or   motivate the specific aim of the study. More constructively, I think it would   be good to discuss such studies as mentioned above, because these factors   such as democracy and corruption will likely prove important in order to   craft effective adaptation strategies in the future for fisheries.

Response:

Thank you very much for your valuable   suggestions. We have already added more discussion on democracy and   corruption as shown in Lines 350-352.

           “On   the other hand, democracy [41] and the corruption [42] affect the fisheries   governance in terms of inequitable policy support and laws enforcement.   Therefore, suppression of corruption and reducing inequality are required

Comment:

Minor changes relates to   grammar and I recommend a thorough reading of the English. See, e.g., page   15, the heading 4.2. should be Factors, with an s, I suppose. Should heading   4 be Discussion rather than Discussions?

Response:

- We have already revised as “4.2 Factors   influencing fishermen’s decision to adapt and expectation on their children   to inherit and be the fisherman” (Lines 281-282)

- We have already revised as “4. Discussion” (Line 258)

Comment:

Some sentences need   rephrasing, such as – on page 3 – “… To answer the main research questions   on:”

Response:

We have already revised as “To answer the main research questions, the following main   points are focused:”   (Lines 103-104)

Round 2

Reviewer 1 Report

The manuscript has been improved in this second round and I feel it is almost ready to be published in CLIMATE.

I only marked some minor errors and few english editing details.

Author Response

Dear Editor and reviewer,

Thank you very much for your kind efforts in reviewing our manuscript.

We greatly appreciate the valuable and constructive comments from the reviewers. Along the lines suggested by the reviewers, we have made comprehensive revisions and have significantly improved the writing of the manuscript. In our responses to the reviewers below, details of the changes we have made in the manuscript can be found orderly.

We hope that the revised manuscript can reach the criteria for publication in Climate.

Yours sincerely,

The authors

Reviewer: 1 (Round 2)

Comment:

I only marked some minor   errors and few english editing details.

Response 1:

    We   have already revised the English error as shown in Lines 39-43.

    “It is predicted that these climate   change impacts will force fisherman villages around coastal areas to adapt   themselves in response to the changing conditions, and in doing so [2], and in doing so, an adaptation strategy needs to be developed based on the data collected   from the fisherman communities, including their knowledge, skills,   experience, and socio-economic context [3]

Comment:

I only marked some minor   errors and few english editing details.

Response 2:

We have already revised the English error as   shown in Line 44.

The fisheries sector provides job opportunities to promote local economy [4]

Comment:

I only marked some minor   errors and few english editing details.

Response 3:

             We have already revised the English error as shown in Lines 70-72.

      These lead to the research questions focus in which are the influence factors for   fisherman’s decision to adapt to climate change, and to expect their   children to inherit and be the fisherman?

Comment:

I only marked some minor   errors and few english editing details.

Response 4:

      We   have already revised the English error as shown in Line 194.

     “…that are likely to increase   also discourage them from expecting their children to be fishermen (Table 2)

Comment:

I only marked some minor   errors and few english editing details.

Response 5:

      We   have already revised the English error as shown in Line 271.

     “…them to cope with the impacts of climate change in the   future.

Reviewer 2 Report

Thanks for the clarifications, which should make the piece better suited for publication.

Author Response

Dear Editor and reviewer,

Thank you very much for your kind efforts in reviewing our manuscript.

Yours sincerely,

The authors
